# Dietary Advice to Support Glycaemic Control and Weight Management in Women with Type 1 Diabetes during Pregnancy and Breastfeeding

**DOI:** 10.3390/nu14224867

**Published:** 2022-11-17

**Authors:** Lene Ringholm, Sidse Kjærhus Nørgaard, Ane Rytter, Peter Damm, Elisabeth Reinhardt Mathiesen

**Affiliations:** 1Center for Pregnant Women with Diabetes, Rigshospitalet, 2100 Copenhagen, Denmark; 2Department of Endocrinology and Metabolism, Rigshospitalet, 2100 Copenhagen, Denmark; 3Department of Clinical Medicine, Faculty of Health and Medical Sciences, University of Copenhagen, 2200 Copenhagen, Denmark; 4The Nutrition Unit, Rigshospitalet, 2100 Copenhagen, Denmark; 5Department of Obstetrics, Rigshospitalet, 2100 Copenhagen, Denmark

**Keywords:** pregnancy, type 1 diabetes, breastfeeding, carbohydrate counting, hypoglycaemia, gestational weight gain, weight retention, ketoacidosis

## Abstract

In women with type 1 diabetes, the risk of adverse pregnancy outcomes, including congenital anomalies, preeclampsia, preterm delivery, foetal overgrowth and perinatal death is 2–4-fold increased compared to the background population. This review provides the present evidence supporting recommendations for the diet during pregnancy and breastfeeding in women with type 1 diabetes. The amount of carbohydrate consumed in a meal is the main dietary factor affecting the postprandial glucose response. Excessive gestational weight gain is emerging as another important risk factor for foetal overgrowth. Dietary advice to promote optimized glycaemic control and appropriate gestational weight gain is therefore important for normal foetal growth and pregnancy outcome. Dietary management should include advice to secure sufficient intake of micro- and macronutrients with a focus on limiting postprandial glucose excursions, preventing hypoglycaemia and promoting appropriate gestational weight gain and weight loss after delivery. Irrespective of pre-pregnancy BMI, a total daily intake of a minimum of 175 g of carbohydrate, mainly from low-glycaemic-index sources such as bread, whole grain, fruits, rice, potatoes, dairy products and pasta, is recommended during pregnancy. These food items are often available at a lower cost than ultra-processed foods, so this dietary advice is likely to be feasible also in women with low socioeconomic status. Individual counselling aiming at consistent timing of three main meals and 2–4 snacks daily, with focus on carbohydrate amount with pragmatic carbohydrate counting, is probably of value to prevent both hypoglycaemia and hyperglycaemia. The recommended gestational weight gain is dependent on maternal pre-pregnancy BMI and is lower when BMI is above 25 kg/m^2^. Daily folic acid supplementation should be initiated before conception and taken during the first 12 gestational weeks to minimize the risk of foetal malformations. Women with type 1 diabetes are encouraged to breastfeed. A total daily intake of a minimum of 210 g of carbohydrate is recommended in the breastfeeding period for all women irrespective of pre-pregnancy BMI to maintain acceptable glycaemic control while avoiding ketoacidosis and hypoglycaemia. During breastfeeding insulin requirements are reported approximately 20% lower than before pregnancy. Women should be encouraged to avoid weight retention after pregnancy in order to reduce the risk of overweight and obesity later in life. In conclusion, pregnant women with type 1 diabetes are recommended to follow the general dietary recommendations for pregnant and breastfeeding women with special emphasis on using carbohydrate counting to secure sufficient intake of carbohydrates and to avoid excessive gestational weight gain and weight retention after pregnancy.

## 1. Introduction

In women with type 1 diabetes, the risk of adverse outcomes, including congenital anomalies, preeclampsia, preterm delivery, foetal overgrowth and perinatal death is 2–4-fold increased compared to the background population and is related to poor glycaemic control [1,2,3,4,5,6]. Foetal overgrowth is a main cause of shoulder dystocia, birth canal lacerations, and operative delivery and may also lead to preterm delivery [4]. Children born overweight to women with type 1 diabetes are at increased risk of obesity, diabetes and cardiovascular disease later in life [7]. Despite several improvements in other clinical outcomes [6,8,9,10,11,12], around 50% of infants are born overweight [5,13,14,15]. Women with type 1 diabetes have lower fertility than women without this condition [16,17].

Glycaemic control is a well-established predictor for foetal overgrowth in women with type 1 diabetes with a strong independent association between HbA1c and foetal growth [4]. In addition, the rise in obesity worldwide has led to more women with type 1 diabetes entering pregnancy with a pre-pregnancy BMI over 25 kg/m^2^ [18,19,20]. The combination of having type 1 diabetes and an elevated BMI in pregnancy has a synergistic effect and is estimated to exacerbate the risk of foetal overgrowth [14]. Excessive gestational weight gain is emerging as an important risk factor for foetal overgrowth and up to 65% of pregnant women with type 1 diabetes have excessive gestational weight gain irrespective of pre-pregnancy BMI [21,22,23]. There is a strong positive association between excessive gestational weight gain and foetal overgrowth, even after adjustment for HbA1c and pre-pregnancy BMI [22,23]. Interventions to limit gestational weight gain are important to reduce the risk of foetal overgrowth and other adverse neonatal outcomes and to thereby improve the health in offspring of women with type 1 diabetes [21,22].

In order to obtain appropriate glycaemic control and gestational weight gain, dietary recommendations during pregnancy are necessary [24,25]. The amount of carbohydrate consumed in a meal is the main dietary factor affecting the postprandial glucose response [18,24]. Glucose is the primary energy substrate for the foetus and is essential for normal foetal brain metabolism and foetal growth [26]. To cover the need for adequate amounts of glucose substrate for a successful pregnancy, a total daily intake of a minimum of 175 g of carbohydrate to all pregnant women irrespective of pre-pregnancy BMI is recommended by National Academy of Medicine (NAM, known as Institute of Medicine) and the American Diabetes Association (ADA) [27,28].

Women with type 1 diabetes are encouraged to breastfeed, and breastfeeding is prevalent in Denmark [29]. Nonetheless, the breastfeeding period is a potentially stressful and demanding period for women with type 1 diabetes [30]. Barriers to optimal diabetes regulation in this life phase include steeply reduced insulin requirements immediately after delivery, adherence to a diet with sufficient carbohydrate intake and frequent blood glucose monitoring while coping with the demands of motherhood [30].

In order to address these clinical challenges during pregnancy and breastfeeding, diabetes management should include a multi-target approach that focuses on adequate dietary advice to secure sufficient intake of micro- and macronutrients, obtain and maintain strict glucose regulation without increasing hypoglycaemia, and promote appropriate gestational weight gain and weight loss after delivery in women with type 1 diabetes [25,28,29].

This review provides the present evidence supporting recommendations for the diet during pregnancy and breastfeeding in women with type 1 diabetes to promote optimized glycaemic control, maternal weight development and foetal growth. The focus is on carbohydrate intake and maternal weight development and is in addition to the general guidelines for healthy pregnant women which will not be covered here. The literature on dietary recommendations during pregnancy and breastfeeding mostly addresses women with type 1 diabetes but can also be applied to pregnant and breastfeeding women with type 2 diabetes and to women with insulin treated gestational diabetes. Reviews covering dietary recommendations for women with gestational diabetes have recently been published [31,32] and will not be covered further in this review.

## 2. Pre-Pregnancy Care

Poor glycaemic control before and during pregnancy is associated with pregnancy complications such as congenital anomalies, preeclampsia, preterm delivery, foetal overgrowth and perinatal death [25]. To reduce the risk of these severe complications, it is important to provide careful pre-pregnancy care with focus on healthy eating and optimizing glycaemic control [33,34]. During pregnancy planning it is thus important to obtain and maintain strict glycaemic control with an HbA1c as close to normal as possible, at least <53 mmol/mol (7.0%) [35,36]. Teaching women with type 1 diabetes how to count carbohydrates and manage weight already before pregnancy can also be recommended. Ideally women should be advised to perform blood glucose monitoring and individual insulin adjustment and how to manage hypoglycaemia already in the pregnancy planning period [37]. Pre-pregnancy care among women with type 1 diabetes including dietary advice may improve pregnancy preparation, facilitate earlier presentation for antenatal care [38] and result in superior glycaemic control during pregnancy compared to women who do not participate in pre-pregnancy care [37,39]. Innovative approaches engaging diabetes caregivers and using virtual technology to raise women’s awareness of and engagement with pre-pregnancy care may be useful to target women with diabetes who are planning pregnancy, especially in disadvantaged groups [39]. One such example is the smartphone application “Pregnant with Diabetes” which provides easily accessible patient education on diet, weight, diabetes management and other clinically important antenatal health information to women with diabetes who are planning pregnancy or who are pregnant [40]. This smartphone application is frequently used, with 75% of women at our centre having downloaded it and half of the women having engaged with it prior to pregnancy [41].

## 3. Glycaemic Control during Pregnancy

Glycaemic control during pregnancy is by the ADA recommended to be as close to normal as is safely possible, ideally HbA1c < 48 mmol/mol (6.5%) in early pregnancy to reduce the risk of congenital anomalies [28] and below 42 mmol/mol (6.0%) during the remaining pregnancy to minimize the risk of preeclampsia, preterm delivery and foetal overgrowth [28]. The Center for Pregnant Women with Diabetes in Copenhagen recommends HbA1c below 38 mmol/mol (5.6%) after 20 gestational weeks [42], which is similar to the upper normal range of HbA1c during pregnancy [43]. However, these HbA1c targets only apply if they can be achieved without a clinically important risk of severe hypoglycaemia, defined as events with symptoms of hypoglycaemia requiring help from another person to actively administer oral carbohydrate or inject glucagon or glucose [44].

The risk of severe hypoglycaemia during pregnancy can be reduced while maintaining good glycaemic control by the use of a multifactorial clinical approach that includes careful timing of meals and adequate dosing of insulin analogues. It is also pivotal to identify women at high risk of severe hypoglycaemia, i.e., women who have a history with severe hypoglycaemia in the year preceding pregnancy, impaired hypoglycaemia awareness, long duration of diabetes, HbA1c < 48 mmol/mol (6.5%) in early pregnancy, fluctuating plasma glucose values, and excessive use of supplementary insulin injections between meals. In contrast, pregnancy-induced nausea and vomiting do not seem to contribute to the risk of severe hypoglycaemia [45,46,47].

To achieve blood glucose levels that are sufficient to prevent adverse outcomes of pregnancy while avoiding severe hypoglycaemia, the ADA suggests the following targets for blood glucose monitoring: fasting glucose levels 3.9–5.3 mmol/L, one-hour postprandial levels of 6.1–7.8 mmol/L and two-hour postprandial levels 5.6–6.7 mmol/L [28].

## 4. Continuous Glucose Monitoring

Continuous glucose monitoring (CGM) as a supplement to or partial substitution for conventional blood glucose monitoring during pregnancy is increasingly used by women with type 1 diabetes, especially following the results of the Continuous Glucose Monitoring in Pregnant Women with Type 1 Diabetes (CONCEPTT) trial [48]. According to the recommendations of an international consensus group, treatment targets for pregnant women using CGM are >70% of time in the range 3.5–7.8 mmol/L and <4% of time <3.5 mmol/L [49]. In young persons aged 10–20 years with type 1 diabetes and a history of severe hypoglycaemia, impaired hypoglycaemia awareness and/or fear of hypoglycaemia, fully subsidized CGM is associated with improved glycaemic control including reduced rates of severe hypoglycaemia [50]. CGM is fully subsidized to all pregnant women with type 1 diabetes in some countries including the UK [51,52], Denmark [53] and Belgium [54]. Wearable non-invasive blood glucose detection technology and transdermal biosensors are emerging as an alternative to conventional blood glucose monitoring [55,56]. The safety, accuracy and efficacy of this technology during pregnancy need to be established in future research.

## 5. Gestational Weight Gain

In 2009, the NAM recommended optimal targets for gestational weight gain based on pre-pregnancy BMI in healthy women and further categorized gestational weight gain as insufficient, appropriate or excessive [57] (Table 1). For women with diabetes, gestational weight gain close to the lower limits of the NAM guidelines or slightly lower seems most appropriate [23].

Gestational weight gain is increasing in women with type 1 diabetes [58], and over half of women with type 1 diabetes have excessive gestational weight gain [18,21,22,23]. Both in healthy women and in women with type 1 diabetes, excessive gestational weight gain increases the risk of foetal overgrowth. The weekly weight gain is highest after 20 gestational weeks compared to early pregnancy [23]. However, in women with type 1 diabetes, excessive gestational weight gain already in first trimester results in an increased risk of foetal overgrowth [22,23] compared to women with appropriate or insufficient gestational weight gain [23].

Gestational weight gain is independently associated with offspring birth weight, also after adjustment for pre-pregnancy BMI and HbA1c in late pregnancy [22,23]. Meanwhile, offspring birth weight is only similar to offspring birth weight in the background population in the small group of women who have inappropriate gestational weight gain according to the NAM recommendations [23]. There is a need for more clinical research on how to obtain appropriate gestational weight gain in women with type 1 diabetes in order to improve the health in offspring of women with type 1 diabetes.

Observations in obese women without diabetes [59] and national Danish guidelines for healthy normal weight women have led to the development of the Copenhagen guidelines on appropriate gestational weight gain in women with diabetes where the recommended gestational weight gain in women with BMI ≥ 30.0 kg/m^2^ is 0–5 kg (Table 2).

## 6. Dietary Advice in Pregnancy

In early pregnancy, women with type 1 diabetes should be referred to a registered dietician who is familiar with the management of type 1 diabetes in pregnancy for medical nutrition therapy, including an individualized meal plan and support to identify and quantify carbohydrates, low-glycaemic-index foods and, as indicated, adjust the amount and timing of carbohydrates in meals (Box 1) [61,62].

Box 1Recommendations for diet and weight management during pregnancy in women with type 1 diabetes.Secure sufficient intake of micro- and macronutrients and, during the first 12 gestational weeks, folic acid supplementation.A meal plan that includes a moderately low carbohydrate content of a minimum of 175 g daily, irrespective of pre-pregnancy BMI.Carbohydrate counting at each meal and snack is helpful to obtain optimized glycaemic control.Carbohydrate intake mainly from low-glycaemic-index sources.Consistent timing of three main meals and 2–4 snacks daily with focus on carbohydrate amount is probably of value to prevent both hypoglycaemia and hyperglycaemia.Gestational weight gain close to the lower limits of the National Academy of Medicine guidelines or according to the Copenhagen guidelines for women with diabetes.

Low carbohydrate diets have become popular and have proven efficient in reducing weight, glycaemic variability, and time spent in hypoglycaemia in non-pregnant persons with type 1 diabetes [63]. However, a low carbohydrate intake might induce lipolysis and ketone production, which in pregnancy is inappropriate because pregnant women are more prone to ketosis than non-pregnant women [64,65], and maternal ketone bodies might have a negative effect on the developing foetal central nervous system [66,67]. Both the maternal brain and the developing foetal brain mainly use glucose as a substrate and the NAM therefore recommends 40 g extra carbohydrate consumption during pregnancy to promote normal foetal growth and brain development [27]. This will be met by a limited carbohydrate intake with a total daily intake of a minimum of 175 g as recommended by the NAM and the ADA to all pregnant women [27,28].

In obese pregnant women without diabetes a lower carbohydrate intake at a moderate level (188 g daily) in late pregnancy has been associated with a lower fat mass in their offspring at birth [68]. Across studies, pregnant women with type 1 diabetes report a mean total carbohydrate intake of approximately 220 g daily [69] with no difference in daily average carbohydrate intake between pre-pregnancy BMI categories [18]. This is 20% less carbohydrate than in healthy pregnant women but still within the target recommended by the NAM and the ADA [27,28].

Generally, across European countries, demographic and socioeconomic factors as age, gender, country of birth and educational level are independently associated with diet quality, as indicated by healthy choices and adherence to dietary guidelines. Economic resources and wealth perception also contribute, but to a lesser extent [70]. In Norwegian adolescents, differences in dietary habits were seen by gender and socioeconomic status. Interestingly, girls reported a higher consumption of healthy foods as vegetables and a lower consumption of unhealthy foods as fast food and soft drinks than boys. Differences were also found in parents’ knowledge of dietary guidelines according to socioeconomic status [71]. During pregnancy, nutritional guidelines are often not being met, even in women with high education, as demonstrated in a multi-centre study from England, Scotland and Ireland where carbohydrate sources of ultra-processed food as sweets, confectionery, biscuits and cakes contributed to almost half of total daily carbohydrate intake while consumption of fibre, fruit and vegetable was inadequate regardless use of multiple daily injections (MDI) or insulin pump therapy [24,72]. Consumption of ultra-processed foods during pregnancy may increase postprandial glucose levels, HbA1c and gestational weight gain [73]. This underscores the importance of advocating a diet including low-glycaemic-index carbohydrate sources while advising against consumption of ultra-processed foods during pregnancy.

Women with diabetes are recommended to follow the general dietary guidelines for pregnancy [33,34]. A Mediterranean diet provides an adequate supply of energy and nutrients during pregnancy and has in healthy pregnant women been associated with reduced risk of gestational diabetes and excessive weight gain without increasing the risk of small for gestational age infants [74,75]. Smoking and alcohol consumption are strongly discouraged during pregnancy [33,34].

## 7. Recommendations on Carbohydrate Intake

The dietary advice of a total daily intake of a minimum of 175 g of carbohydrate [27,28] can be met by consuming 150 g of carbohydrate from the main carbohydrate sources such as bread, whole grain, fruits, rice, potatoes, dairy products and pasta and 25 g from vegetables, almonds or other minor carbohydrate sources [61]. These food items are often available at a lower cost than ultra-processed foods, so this dietary advice is likely to be feasible also in women with low socioeconomic status. At our centre, we primarily recommend a daily intake of 20, 50 and 50 g carbohydrate at breakfast, lunch and dinner, respectively, and 10 g at three snacks [40]. Consistent timing of all meals and snacks is likely to be of value to prevent both hypoglycaemia and hyperglycaemia in pregnant women with type 1 diabetes [61]. Blood glucose levels often increase to high levels after breakfast, whereas a low-carbohydrate breakfast may lower postprandial glucose excursions after breakfast [76]. We therefore recommend a relatively low amount of carbohydrates in the breakfast [61]. The diet should include mainly low-glycaemic-index carbohydrates sources that are slowly absorbed, while single large meals and ultra-processed food as sweets, confectionery, biscuits and cakes should be kept to a minimum or, preferably, left out of the diet to avoid large postprandial glucose excursions (Box 1) [28,61].

Carbohydrate counting is an integral component of modern dietary management and a helpful pragmatic strategy to match the dose of mealtime insulin to the amount of carbohydrate intake at meals. Regular review of recorded food intake may also be helpful to evaluate carbohydrate intake and to help women avoid excessive gestational weight gain based on the weekly weight development [62,77,78]. Women with type 1 diabetes who use carbohydrate counting obtain better HbA1c in early pregnancy compared to women who do not count carbohydrates [79]. A positive association between HbA1c levels and the quantity of carbohydrates consumed in both early pregnancy [79] and late pregnancy has been observed irrespective of BMI, gestational weight gain, insulin dose or energy intake [18]. Smartphone applications to ease carbohydrate counting are available in many languages and at least a pragmatic use of carbohydrate counting is often obtainable.

The NAM recommends healthy women to obtain 45–65% of total daily energy intake from carbohydrate [27]. We have previously suggested a moderately low carbohydrate diet with 40% carbohydrates for pregnant women with type 1 diabetes [61].

Recommendations for diet and weight management during pregnancy are given in Box 1.

## 8. Folic Acid

Daily folic acid supplementation should be initiated before conception and taken during the first 12 gestational weeks to reduce the risk of congenital anomalies in the offspring of women with diabetes and the intake should preferably begin while planning pregnancy [80]. There is no consensus on the dose of folic acid, and recommended doses range from 400 μg/day to 5 mg/day [28,81].

## 9. Breastfeeding in Women with Type 1 Diabetes

Full breastfeeding for the first 4–6 months of life is recommended by the World Health Organization [82]. Both the woman and her newborn may benefit from long-term breastfeeding. In the woman it may promote weight loss and prevent weight retention after pregnancy [83,84]. In the offspring, breastfeeding may protect against future development of obesity [85,86,87,88], type 1 [89,90,91] and type 2 diabetes [92,93].

In women with type 1 diabetes, previous experience with breastfeeding, higher educational level, and number of breastfeeds within 24 h after delivery are positively associated with a longer breastfeeding period while higher pre-pregnancy BMI and smoking are negatively associated with breastfeeding [94,95,96].

## 10. Dietary Advice during Breastfeeding

Carbohydrate in human milk is almost exclusively lactose composed of glucose and galactose [27]. To secure sufficient carbohydrate intake for the milk production, the NAM recommends a daily total of a minimum of 210 g carbohydrate to all breastfeeding women irrespective of pre-pregnancy BMI [27]. For a balanced diet, the main carbohydrate sources can be bread, fruits, rice, potatoes, dairy products and pasta [29].

The additional energy costs for exclusive breastfeeding from delivery until six months are approximately 620–670 kilocalories (kcal) daily [83,84]. Gradual weight loss will subsidize this additional energy cost by approximately 170 kcal daily from energy stores accumulated during pregnancy [83], so the net increase in energy requirement during the first six months of exclusive breastfeeding is approximately 450–500 kcal daily [83,84]. Women who are breastfeeding are recommended to consume a minimum of 1800 kcal daily [97].

### Ketoacidosis

Healthy breastfeeding women following a diet with low carbohydrate content may be depleted of the hepatic glycogen stores and develop a relative insulin deficiency [98,99,100,101]. The cerebral demand for energy is subsequently met by hepatic gluconeogenesis and mobilization of free fatty acids from adipose tissue with formation of ketone bodies as an alternative energy substrate [98,100,101]. This easily creates a negative energy balance with nutritional ketosis and ketoacidosis [102], so-called lactation ketoacidosis [29]. There are several case reports on lactation ketoacidosis in women without diabetes [98,99,100,101,102,103,104,105,106,107,108,109,110,111,112,113,114,115] where low or no carbohydrate intake was the main contributor, while rehydration and resumption of a normal diet resulted in complete resolution of the symptoms and the ketoacidosis [98,99,100,101,102,104,105,106,107,108,109,110,111,112,113,114,115]. Searching the literature, including recent clinical studies in breastfeeding women with type 1 diabetes [30,94,116,117,118,119,120], no cases of ketoacidosis during breastfeeding in women with type 1 diabetes have been published. Nonetheless, women with type 1 diabetes should be encouraged to follow the recommendation on a daily carbohydrate intake of a minimum of 210 g during breastfeeding [27] to secure enough glucose for an adequate volume of breast milk, to avoid hypoglycaemia and to prevent lactation ketoacidosis [119,121].

## 11. Glycaemic Control during Breastfeeding

In all women with type 1 diabetes, whether intending to breastfeed or not, immediately after delivery the insulin requirements are approximately 30% lower than before pregnancy owing to loss of placental growth hormone, insulin-like growth factor-I and other placental hormonal influence upon delivery [116,119,122]. Over the next weeks and months after delivery insulin requirements in breastfeeding women are reported approximately 20% lower than before pregnancy with wide individual variation regardless use of MDI therapy or insulin pump therapy [29].

HbA1c may not be an optimal marker of glycaemic control within the first few months after delivery owing to a physiological fall in HbA1c levels from early to late pregnancy [43]. Furthermore, even a normal blood loss during delivery and puerperium may be followed by a low HbA1c within the first one to two months after delivery [28].

Women with type 1 diabetes who breastfeed may experience less glucose variability than those who do not breastfeed [117]. In a prospective study on CGM use for six days at one, two and six months after delivery breastfeeding women with type 1 diabetes, who were advised to consume a minimum of 210 g carbohydrate daily, spent more time in the target range 4.0–10.0 mmol/L compared to non-pregnant, non-breastfeeding control women with type 1 diabetes [119]. At six months after delivery, HbA1c was lower in the breastfeeding women than in the control women [119], but generally HbA1c may deteriorate 6–18 months after delivery with wide individual variations [123]. In a secondary analysis, 13 women on insulin pump therapy spent ≥71% of time in the target range 4.0–10.0 mmol/L, both at night and for 24 h [118]. In a retrospective study on 44 women with type 1 diabetes, where 80% fully breastfed [120], the treatment targets were fasting glucose levels <7.2 mmol/L and postprandial levels <10.0 mmol/L.

Overall, during the breastfeeding period it is probably wise to aim for preprandial glucose levels 4.0–7.0 mmol/L and at all other times 4.0–10.0 mmol/L. According to international consensus, CGM targets for breastfeeding women are the same as in other non-pregnant persons with diabetes (>70% of time in the range 3.9–10.0 mmol/L and <4% of time <3.9 mmol/L) [49]. Up to 2–3 episodes per week with mild hypoglycaemia (events with symptoms familiar to the woman as hypoglycaemia and managed by herself [124]) are tolerated [119].

### Hypoglycaemia during Breastfeeding

In women with normal glucose tolerance, suckling does not affect glucose profiles, either in the fasting or non-fasting state [125,126]. Likewise, breastfeeding does not usually induce hypoglycaemia in women with type 1 diabetes [29]. In 438 recorded breastfeeding episodes at night without carbohydrate intake among 26 breastfeeding women with type 1 diabetes, the vast majority (>95%) of breastfeeds were not followed by hypoglycaemia within the next three hours [119], and in a small study of eight women with type 1 diabetes, glucose levels remained over 4.0 mmol/L after the majority of breastfeeds [117].

In a prospective study including 26 breastfeeding women with type 1 diabetes who were recommended a daily carbohydrate intake of 210 g according to the NAM recommendation [27], CGM for six days was applied at one, two and six months after delivery. Insulin doses were reduced immediately after pregnancy and after each CGM period, when appropriate, resulting in median 5.1% (range 0–20%) amount of time in the hypoglycaemic range (CGM < 4.0 mmol/L) regardless use of MDI or insulin pump therapy [118,119].

The prevalence of severe hypoglycaemia within four months after pregnancy in women with type 1 diabetes was similar in breastfeeding and formula feeding women, respectively, (11% versus 13%) in one recent clinical study [94].

Historically, there have been concerns that women with type 1 diabetes may be troubled with hypoglycaemia shortly after each breastfeed [127,128,129]. The risk of hypoglycaemia has been considered to be greatest in the first two-four weeks after delivery, although this is only based on small studies in breastfeeding women with type 1 diabetes [30,116,130]. Previously, women were often advised to routinely consume carbohydrates at night-time breastfeeds out of concern for hypoglycaemia following breastfeeding [119,131]. However, this recommendation is probably obsolete in women consuming an appropriate daily amount of carbohydrate with properly reduced insulin doses.

Recommendations for diet and weight management during breastfeeding are given in Box 2.

Box 2Recommendations for diet and weight management during breastfeeding in women with type 1 diabetes.Promotion of breastfeeding has high priority because of its advantages for both women and infants.Secure sufficient intake of micro- and macronutrients.A meal plan that includes a daily intake of a minimum of 210 g of carbohydrate, regardless pre-pregnancy BMI.Carbohydrate intake mainly from low-glycaemic-index sources.Immediately after delivery insulin requirements are approximately 30% lower than before pregnancy.Over the next weeks and months insulin requirements are approximately 20% lower than before pregnancy.A snack at each breastfeed at night-time is not necessary when women consume an appropriate daily amount of carbohydrate and their insulin doses are appropriate.Aim for prevention of maternal weight retention after pregnancy.

## 12. Weight Retention after Pregnancy

Women who have excessive gestational weight gain have a greater risk of being overweight and obese later in life [132,133]. However, when breastfeeding and adequate maternal nutritional therapy are established, it is possible to gradually lose weight without harming the infant [97]. Weight changes in the breastfeeding period vary and depend on gestational weight gain as well as the pre-pregnancy BMI, breastfeeding duration and pattern, physical activity and maternal age [83,134].

Factors with a positive impact on weight loss and achievement of the pre-pregnancy weight include appropriate gestational weight gain and appropriate total daily energy intake after delivery, including adequate quantity and quality of carbohydrates [60,119]. In women with type 1 diabetes, adequate reduction in the insulin dose immediately after pregnancy and when indicated is also important to limit the need for extra carbohydrate intake for hypoglycaemia [119].

The literature on the influence of breastfeeding on weight loss after pregnancy in healthy women is not conclusive. Less weight retention in breastfeeding women compared to formula feeding women has been described in some [134,135], but not all [136] studies, and overall, solid long-term data on weight development after pregnancy is difficult to collect due to high drop-out rates [123,137].

In women with type 1 diabetes, weight retention 4 months after delivery was on average 0.8 kg lower in breastfeeding women compared to formula feeding women, although this difference was not statistically significant [94]. In a small study on 26 breastfeeding women with type 1 diabetes, weight retention was less than five kilograms already two months after delivery [119]. The recommendation to consume at least 210 g of carbohydrate daily during breastfeeding therefore does not seem to preclude women with type 1 diabetes from weight loss or achievement of their pre-pregnancy weight [119]. However, in a Polish study on women with type 1 diabetes, body weight 20 months after delivery were greater than before pregnancy, by 2.5 kg, but data on breastfeeding were not available and study results may be affected by drop-out rates >37% after delivery [123].

## 13. Conclusions

Management of type 1 diabetes during pregnancy includes careful counselling about adequate diet with focus on appropriate amounts and quality of carbohydrate, timing of meals and avoidance of both hypoglycaemia and hyperglycaemia. The recommended food items with high carbohydrate content and low glycaemic index, such as bread, whole grain, fruits, rice, potatoes, dairy products and pasta, are often available at a lower cost than ultra-processed foods, so this dietary advice is likely to be feasible also in women with low socioeconomic status. Appropriate gestational weight gain should be promoted.

Women with type 1 diabetes are encouraged to breastfeed. Appropriate carbohydrate intake and reduction of insulin doses compared to pre-pregnancy doses are important to maintain acceptable glycaemic control without hypoglycaemia in the breastfeeding period. Weight loss after pregnancy is important to prevent overweight and obesity later in life.

The recommendations from the Center for Pregnant Women with Diabetes in Copenhagen on dietary advice and weight management during pregnancy and the breastfeeding period are summarized in the smartphone application “Pregnant with Diabetes”, which is available for pregnant women and their caregivers worldwide [40].

### Key Points on Dietary Advice to Women with Type 1 Diabetes during Pregnancy and Breastfeeding

Sufficient intake of micro- and macronutrients and, during the first 12 gestational weeks, folic acid supplementation.

Careful, individual counselling about appropriate diet aiming to achieve optimized glycaemic control and promote appropriate weight development during pregnancy and breastfeeding while avoiding hypoglycaemia.

Limitation of carbohydrate intake but a minimum of 175 g daily during pregnancy, mainly from low-glycaemic-index carbohydrates sources, irrespective of pre-pregnancy BMI.

Recommended gestational weight gain is dependent on maternal pre-pregnancy BMI and is lower when BMI is above 25 kg/m^2^.

Three main meals and 2–4 snacks daily are recommended during pregnancy and in the breastfeeding period.

A total daily intake of a minimum of 210 g of carbohydrate is recommended during breastfeeding, irrespective of pre-pregnancy BMI.

Aim to avoid weight retention after pregnancy.

## Figures and Tables

**Table 1 nutrients-14-04867-t001:** Recommendations for healthy women: Total gestational weight gain according to National Academy of Medicine guidelines 2009, based on pre-pregnancy BMI including a daily intake of a minimum of 175 g of carbohydrate irrespective of pre-pregnancy BMI [27,57].

Pre-Pregnancy BMI (kg/m^2^)	Recommended Total Gestational Weight Gain in Healthy Pregnant Women According to National Academy of Medicine 2009 (kg)
≤18.5 Underweight	12.5–18.0
18.5–24.9 Normal weight	11.5–16.0
25.0–29.9 Overweight	7.0–11.5
≥30.0 Obese	5.0–9.0

**Table 2 nutrients-14-04867-t002:** Recommendations for pregnant women with diabetes: The Copenhagen guidelines on total gestational weight gain for women with diabetes, based on pre-pregnancy BMI [60] including a daily intake of a minimum of 175 g of carbohydrate, mainly from low-glycaemic-index carbohydrates sources, irrespective of pre-pregnancy BMI [61].

Pre-Pregnancy BMI (kg/m^2^)	Recommended Total Gestational Weight Gain in Women with Diabetes According to the Copenhagen Guidelines (kg)
18.5–24.9 Normal weight	10.0–15.0
25.0–29.9 Overweight	5.0–8.0
≥30.0 Obese	0–5.0

## Data Availability

Not applicable.

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
