# Peer review of "Dietary Advice to Support Glycaemic Control and Weight Management in Women with Type 1 Diabetes during Pregnancy and Breastfeeding"

_nutrients, 2022, doi:10.3390/nu14224867_

Round 1

Reviewer 1 Report

Abstract

line 12: Re write the first sentence to clarify the gap in the research market covered by this short review

Introduction

line 45: A bit more expansion here in order to provide a broader context for the subject of this paper? 

Can you emphasise the parental inter-generational context for  the birth of diabetic-prone overweight offspring. This  is a  long-standing cultural issue.   The  problem has persisted since the abundance of refined foods and growth of super markets  in the mid 1970s in many countries, particularly in the USA. The problem is related to national food policy and is normally associated with socioeconomic status.  I think it relevant to mention these wider issues. Does dietary mismanagement exist  across all socio economic classes in al countries?  Interesting if it does e.g. you have a comparatively egalitarian society in Denmark. 

Is there room for mentioning the effects of government interventions for monitoring salt and sugar content of refined foods. Are there yet papers out on the effects of these interventions on the incidence of type 1 diabetes anywhere? How do pregnant women fare in the developing world in indigenous cultures? Any information - I know that the plague of diabetes spreads with the increased uptake of refined diets in the developing world as well. There may be literature on indigenous diets and blood sugar levels? You briefly  mention the benefits of the  Mediterranean diet.  I am trying to think of ways of expanding the novelty in the article. 

https://hypertension.zaslavsky.com.ua/index.php/journal/article/view/325

There are success stories for managing diabetes if continuous glucose monitoring is free:

https://diabetesjournals.org/care/article/45/2/391/139012/Universal-Subsidized-Continuous-Glucose-Monitoring

line 93: I think it relevant to briefly mention the research into the development of portable transdermal sensors   sensors  for continual circulatory  glucose monitoring in diabetics. Removing the requirement for a blood sample to monitor glucose should improve patient compliance. 

https://www.mdpi.com/1424-8220/20/23/6925

https://www.sciencedirect.com/science/article/pii/S0958166921001518

lines 275  to 470

General comment: 

How does this information move the subject on or were is the evidence for a short review on the content? I am not an expert in this field but the content appears to be largely well established and factual - where is the novel context? A novel way of linking topics in order to provide deeper insight. 

Paragraphs 6,7 and 8 are very brief. If you have covered all the relevant literature on pregnant women with type 1 diabetes, perhaps illustrated and widen and deepen your content by providing information dietary management and glucose monitoring in children - a second vulnerable group and the inheritors of maternal dietary mismanagement. 

Can the review be widened to policy issues - socioeconomic status  and interventions where continual glucose monitoring has been successful. 

The content generally is too  'light weight' in these sections - as it stands. It needs a novel context. 

Conclusion

Bullet points do not belong in articles - a clinical report maybe but not an article where novel ideas are synthesised. 

Author Response

Reply to Reviewer 1:

line 12: Re write the first sentence to clarify the gap in the research market covered by this short review

Answer:

Thank you for this suggestion.

We have changed the first part of the abstract to the following:

“In women with type 1 diabetes the risk of adverse pregnancy outcomes, including congenital anomalies, preeclampsia, preterm delivery, fetal overgrowth and perinatal death is 2-4-fold increased compared to the background population. The amount of carbohydrate consumed in a meal is the main dietary factor affecting the postprandial glucose response. Excessive gestational weight gain is emerging as another important risk factor for fetal overgrowth. Dietary advice to promote optimized glycaemic control and appropriate gestational weight gain is therefore important for normal fetal growth and pregnancy outcome. This review provides the present evidence supporting recommendations for the diet during pregnancy and breastfeeding in women with type 1 diabetes.”

We have partly rewritten the remaining part of the abstract:

“Dietary management should include advice to secure sufficient intake of micro- and macronutrients with focus on limiting postprandial glucose excursions, preventing hypoglycaemia, and promoting appropriate gestational weight gain and weight loss after delivery. Irrespective of pre-pregnancy BMI, limitation of total daily carbohydrate intake but including at least 175 grams, mainly from low glycaemic index sources as bread, whole grain, fruits, rice, potatoes, dairy products and pasta is recommended during pregnancy. These food items are often available at lower cost than ultra-processed foods, so this dietary advice is likely to be feasible also in women with low socioeconomic status. Individual counselling aiming at consistent timing of three main meals and 2-4 snacks daily, with focus on carbohydrate amount with pragmatic carbohydrate counting, is probably of value to prevent both hypoglycaemia and hyperglycaemia. The recommended gestational weight gain is dependent on maternal pre-pregnancy BMI and is lower when BMI is above 25 kg/m2. Daily folic acid supplementation should be initiated before conception and taken during the first 12 gestational weeks to minimize the risk of fetal malformations. Women with type 1 diabetes are encouraged to breastfeed. A total daily intake of minimum 210 grams of carbohydrate is recommended in the breastfeeding period for all women irrespective of pre-pregnancy BMI to maintain acceptable glycaemic control while avoiding ketoacidosis and hypoglycaemia. During breastfeeding insulin requirements are reported approximately 20% lower than before pregnancy. Women should be encouraged to avoid weight retention after pregnancy in order to reduce the risk of overweight and obesity later in life.”

In conclusion, pregnant women with type 1 diabetes are recommended to follow the general dietary recommendations for pregnant and breastfeeding women with special emphasis on using carbohydrate counting to secure sufficient intake of carbohydrates and to avoid excessive gestational weigth gain and weight retention after pregnancy.

Introduction

line 45: A bit more expansion here in order to provide a broader context for the subject of this paper? 

Answer:

Thank you for this suggestion. We now emphasize that excessive gestational weight gain is emerging as an important risk factor for fetal overgrowth and up to 65% of pregnant women with type 1 diabetes have excessive gestational weight gain irrespective of pre-pregnancy BMI. Interventions to limit gestational weight gain are important to reduce the risk of fetal overgrowth and other adverse neonatal outcomes.

We have expanded the following paragraph in the Introduction page 5:

“Glycaemic control is a well-established predictor for fetal overgrowth …”

“…the rise in obesity worldwide has led to more women with type 1 diabetes entering pregnancy with a pre-pregnancy BMI over 25 kg/m2 (18-20). The combination of having type 1 diabetes and an elevated BMI in pregnancy has a synergistic effect and is estimated to exacerbate the risk of fetal overgrowth. Excessive gestational weight gain is emerging as an important risk factor for fetal overgrowth and up to 65% of pregnant women with type 1 diabetes have excessive gestational weight gain irrespective of pre-pregnancy BMI. There is a strong positive association between excessive gestational weight gain and fetal overgrowth, even after adjustment for HbA1c and pre-pregnancy BMI. Interventions to limit gestational weight gain are important to reduce the risk of fetal overgrowth and other adverse neonatal outcomes and to thereby improve the health in offspring of women with type 1 diabetes.”

On page 10 we have added:

“There is a need for more clinical research on how to obtain appropriate gestational weight gain in women with type 1 diabetes in order to improve the health in offspring of women with type 1 diabetes.”

Can you emphasise the parental inter-generational context for  the birth of diabetic-prone overweight offspring. This  is a  long-standing cultural issue.   The  problem has persisted since the abundance of refined foods and growth of super markets  in the mid 1970s in many countries, particularly in the USA. The problem is related to national food policy and is normally associated with socioeconomic status.  I think it relevant to mention these wider issues. Does dietary mismanagement exist  across all socio economic classes in al countries?  Interesting if it does e.g. you have a comparatively egalitarian society in Denmark. 

Answer:

Generally ultra-processed food should be kept to a minimum or left out of the diet in pregnant women with type 1 diabetes. There are significant differences in dietary habits according to gender and socioeconomic status across European countries. However, during pregnancy nutritional guidelines are often not being met, even in women with high education. The dietary advice to pregnant women with diabetes is a limited carbohydrate intake with a minimum of 175 grams daily. This minimum intake can be met by consuming for example mainly bread, whole grain, fruits, rice, potatoes, dairy products and pasta. These food items are often available at lower cost than ultra-processed foods, so this dietary advice is likely to be feasible in many countries.

On page 11 we have added:

“In obese pregnant women without diabetes a lower carbohydrate intake at a moderate level (188 grams daily) in late pregnancy was associated with a lower fat mass in their offspring at birth.”

On page 12 we have added:

“Generally, across European countries, demographic and socioeconomic factors as age, gender, country of birth and educational level are independently associated with diet quality, as indicated by healthy choices and adherence to dietary guidelines. Economic resources and wealth perception also contribute, but to a lesser extent. In Norwegian adolescents differences in dietary habits were seen by gender and socioeconomic status. Interestingly, girls reported a higher consumption of healthy foods as vegetables and a lower consumption of unhealthy foods as fast food and soft drinks than boys. Differences were also found in parents’ knowledge of dietary guidelines according to socioeconomic status. During pregnancy, nutritional guidelines are often not being met, even in women with high education, as demonstrated in a multi-center study from England, Scotland and Ireland where carbohydrate sources of ultra-processed food as sweets, confectionery, biscuits and cakes contributed to almost half of total daily carbohydrate intake while consumption of fibre, fruit and vegetable was inadequate regardless use of multiple daily injections (MDI) or insulin pump therapy. Consumption of ultra-processed foods during pregnancy may increase postprandial glucose levels, HbA1c and gestational weight gain. This underscores the importance of advocating a diet including low glycemic index carbohydrate sources while advising against consumption of ultra-processed foods during pregnancy.”

On page 13 we have added:

These food items are often available at lower cost than ultra-processed foods, so this dietary advice is likely to be feasible in many countries.

On page 13 we now emphasize:

“…single large meals and ultra-processed food as sweets, confectionery, biscuits and cakes should be kept to a minimum or, preferably, left out of the diet …”

In Conclusion on page 20 we have added:

The recommended food items with high carbohydrate content and low glycaemic index as bread, whole grain, fruits, rice, potatoes, dairy products and pasta, are often available at lower cost than ultra-processed foods, so this dietary advice is likely to be feasible also in women with low socioeconomic status.

Is there room for mentioning the effects of government interventions for monitoring salt and sugar content of refined foods. Are there yet papers out on the effects of these interventions on the incidence of type 1 diabetes anywhere? How do pregnant women fare in the developing world in indigenous cultures? Any information - I know that the plague of diabetes spreads with the increased uptake of refined diets in the developing world as well. There may be literature on indigenous diets and blood sugar levels? You briefly  mention the benefits of the  Mediterranean diet.  I am trying to think of ways of expanding the novelty in the article. 

https://hypertension.zaslavsky.com.ua/index.php/journal/article/view/325

Answer:

Thank you for this interesting comment. The possible benefit of a Mediterranean diet is mentioned on page 12 and is covered in the recent reviews dealing with diet to women with gestational diabetes which we refer to on page 7. On page 12-13 we also mention that ultrarefined food is not recommended for pregnant women with diabetes. Despite the fact that the areas raised by the reviewer certainly are of interest, we would like to keep our review focused on diabetes and pregnancy and have therefore chosen not to go further into debt with the issue of the importance of monitoring the amount of salt and sugar in refined food for the general population.

There are success stories for managing diabetes if continuous glucose monitoring is free:

https://diabetesjournals.org/care/article/45/2/391/139012/Universal-Subsidized-Continuous-Glucose-Monitoring

Answer:

Thank you for this suggestion. We now quote the suggested reference. We have also added that use of CGM during pregnancy is fully subsidized in some countries.

Continuous glucose monitoring is now presented in a separate section on page 9.

We have added the following to the text on page 9:

“In young persons aged 10-20 years with type 1 diabetes and a history of severe hypoglycaemia, impaired hypoglycaemia awareness and/or fear of hypoglycaemia, fully subsidized CGM is associated with improved glycaemic control including reduced rates of severe hypoglycaemia. CGM is fully subsidized to all pregnant women with type 1 diabetes in some countries including the UK, Denmark and Belgium.”

line 93: I think it relevant to briefly mention the research into the development of portable transdermal sensors   sensors  for continual circulatory  glucose monitoring in diabetics. Removing the requirement for a blood sample to monitor glucose should improve patient compliance. 

https://www.mdpi.com/1424-8220/20/23/6925

https://www.sciencedirect.com/science/article/pii/S0958166921001518

Answer:

Thank you for this suggestion. We acknowledge the research and development of this non-invasive blood glucose detection technology and transdermal biosensors which our group has been taking part in over the last many years. Their safety, accuracy and efficacy during pregnancy should be established in future research.

On page 9 we have added:

Wearable non-invasive blood glucose detection technology and transdermal biosensors are emerging as an alternative to conventional blood glucose monitoring. The safety, accuracy and efficacy of this technology during pregnancy need to be established in future research.

lines 275  to 470

General comment: 

How does this information move the subject on or were is the evidence for a short review on the content? I am not an expert in this field but the content appears to be largely well established and factual - where is the novel context? A novel way of linking topics in order to provide deeper insight. 

Answer:

Thank you for this comment. The novelty of this review is that we address the need for special emphasis on using carbohydrate counting to secure sufficient intake of carbohydrates and to avoid excessive gestational weight gain and weight retention after pregnancy.

Even though this review includes original papers evaluating this main topic published within the last 10 years by us and other groups, no review has until now collected this evidence in a well-written and easy-to-read way.

We have now added an introduction and conclusion in our abstract to emphasise the novelty of the review.

Paragraphs 6,7 and 8 are very brief. If you have covered all the relevant literature on pregnant women with type 1 diabetes, perhaps illustrated and widen and deepen your content by providing information dietary management and glucose monitoring in children - a second vulnerable group and the inheritors of maternal dietary mismanagement. 

Answer:

Clinical research in type 1 diabetes during the breastfeeding period is generally scarce and we believe we have covered the relevant literature. Information on dietary management and glucose monitoring in children is beyond the scope of this review and our clinical area. We find that expanding the topic of this review to also cover dietary management and glucose monitoring in children will be too broad and make the review too long. 

Can the review be widened to policy issues - socioeconomic status  and interventions where continual glucose monitoring has been successful. 

The content generally is too  'light weight' in these sections - as it stands. It needs a novel context. 

Answer:

We have sought to deal more with these topics both in the abstract and in the main text.

On page 12 we have added:

“Generally across European countries, demographic and socioeconomic factors as age, gender, country of birth and educational level are independently associated with diet quality, as indicated by healthy choices and adherence to dietary guidelines. Economic resources and wealth perception also contribute, but to a lesser extent. In Norwegian adolescents differences in dietary habits were seen by gender and socioeconomic status. Interestingly, girls reported a higher consumption of healthy foods as vegetables and a lower consumption of unhealthy foods as fast food and soft drinks than boys. Differences were also found in parents’ knowledge of dietary guidelines according to socioeconomic status. During pregnancy, nutritional guidelines are often not being met, even in women with high education…”,

On page 9 we have added:

“In young persons aged 10-20 years with type 1 diabetes and a history of severe hypoglycaemia, impaired hypoglycaemia awareness and/or fear of hypoglycaemia, fully subsidized CGM is associated with improved glycaemic control including reduced rates of severe hypoglycaemia. CGM is fully subsidized to all pregnant women with type 1 diabetes in some countries including the UK, Denmark and Belgium.”

Conclusion

Bullet points do not belong in articles - a clinical report maybe but not an article where novel ideas are synthesised. 

Answer:

The manuscript includes Box 1 and Box 2 which both include bullet points. We refer to these two boxes in the manuscript. The conclusion of the manuscript holds no bullet points, but in the manuscript version provided for review the conclusion is mistakenly merged with Box 1 and Box 2.

In the revised version we have highlighted the legends to all boxes, tables and figure and kindly ask the journal not to merge the main text of the manuscript (including the conclusion) with boxes, tables or figure.

Reviewer 2 Report

The following are some questions and comments from this review.

Is the percentage of pregnant women with type 1 diabetes lower than that of non-pregnant women?

Tables 1 and 2 show weight, but could carbohydrate intake be included in the tables?

This review is about type 1 diabetes, but I would have liked to see a background section on the differences in blood glucose and dietary management of women with pre-pregnancy type 2 diabetes and women with gestational diabetes.

I believe that women with type 1 diabetes should be able to self-manage their diet and blood glucose levels even before pregnancy.
This review also describes things to work on before pregnancy, such as folic acid supplementation.
I would have liked to see a sentence between 1 and 2 about pre-pregnancy.

Author Response

Reply to Reviewer 2:

Is the percentage of pregnant women with type 1 diabetes lower than that of non-pregnant women?

Answer:

Women with type 1 diabetes have lower fertility than persons without this condition

In Introduction page 5 we have added:

Women with type 1 diabetes have lower fertility than persons without this condition.

Tables 1 and 2 show weight, but could carbohydrate intake be included in the tables?

Answer:

Thank you for this comment. We have changed the legends of the tables according to suggestions.

Legend to Table 1: Recommendations for women without diabetes: Total gestational weight gain according to National Academy of Medicine guidelines 2009, based on pre-pregnancy BMI  including a daily intake of minimum 175 grams of carbohydrate irrespective of pre-pregnancy BMI.

Legend to Table 2. Recommendations for pregnant women with diabetes: The Copenhagen guidelines on total gestational weight gain for women with diabetes, based on pre-pregnancy BMI  including a daily intake of minimum 175 grams of carbohydrate, mainly from low glycaemic index carbohydrates sources, irrespective of pre-pregnancy BMI

This review is about type 1 diabetes, but I would have liked to see a background section on the differences in blood glucose and dietary management of women with pre-pregnancy type 2 diabetes and women with gestational diabetes.

Answer:

The literature on dietary recommendations during pregnancy and breastfeeding mostly addresses women with type 1 diabetes, but can also be applied to pregnant and breastfeeding women with type 2 diabetes. Dietary recommendations for women with gestational diabetes have recently been published and will not be covered in this review.

In Introduction page 6-7 we have added:

“The literature on dietary recommendations during pregnancy and breastfeeding mostly addresses women with type 1 diabetes, but can also be applied to pregnant and breastfeeding women with type 2 diabetes and to women with insulin treated gestational diabetes. Reviews covering dietary recommendations for women with gestational diabetes have recently been published and will not be covered further in this review.”

I believe that women with type 1 diabetes should be able to self-manage their diet and blood glucose levels even before pregnancy.

Answer:

We agree that optimally women with type 1 diabetes should self-manage their diabetes and diet already before pregnancy.

We have added this paragraph on page 7-8:

“Poor glycaemic control before and during pregnancy is associated with pregnancy complications such as congenital malformations, preeclampsia, preterm delivery, fetal overgrowth and perinatal death. To reduce the risk of these severe complications, it is important to provide careful pre-pregnancy care with focus on healthy eating and optimizing glycaemic control. During pregnancy planning it is thus important to obtain and maintain strict glycaemic control with an HbA1c as close to normal as possible, at least <53 mmol/mol (7.0%). Teaching women with type 1 diabetes how to count carbohydrates and manage weight already before pregnancy can also be recommended. Ideally women should be advised to perform blood glucose monitoring and individual insulin adjustment and how to manage hypoglycaemia already in the pregnancy planning period. Pre-pregnancy care among women with type 1 diabetes including dietary advice may improve pregnancy preparation, facilitate earlier presentation for antenatal care and result in superior glycaemic control during pregnancy.”

This review also describes things to work on before pregnancy, such as folic acid supplementation.
I would have liked to see a sentence between 1 and 2 about pre-pregnancy.

Answer:

Thank you for this comment.

We have added a paragraph on pre-pregnancy care including glycaemic control on page 7-8:

“Poor glycaemic control before and during pregnancy is associated with pregnancy complications such as congenital malformations, preeclampsia, preterm delivery, fetal overgrowth and perinatal death. To reduce the risk of these severe complications, it is important to provide careful preconception care with focus on healthy eating and optimizing glycaemic control. During pregnancy planning it is thus important to obtain and maintain strict glycaemic control with an HbA1c as close to normal as possible, at least <53 mmol/mol (7.0%). Teaching women with type 1 diabetes how to count carbohydrates and manage weight already before pregnancy can also be recommended. Ideally women should be advised to perform blood glucose monitoring and individual insulin adjustment and how to manage hypoglycaemia already in the pregnancy planning period. Pre-pregnancy care among women with type 1 diabetes including dietary advice may improve pregnancy preparation, facilitate earlier presentation for antenatal care and result in superior glycaemic control during pregnancy.”

On page 14 we write the following about folic acid supplementation before and during pregnancy:

“Daily folic acid supplementation should be initiated before conception and taken during the first 12 gestational weeks to reduce the risk of congenital anomalies in the offspring of women with diabetes and the intake should preferably begin while planning pregnancy. There is no consensus on the dose of folic acid, and recommended doses range from 400 μg/day to 5 mg/day.”
